# Motor Neuroplastic Effects of a Novel Paired Stimulation Technology in an Incomplete Spinal Cord Injury Animal Model

**DOI:** 10.3390/ijms23169447

**Published:** 2022-08-21

**Authors:** Muhammad Adeel, Bor-Shing Lin, Hung-Chou Chen, Chien-Hung Lai, Jian-Chiun Liou, Chun-Wei Wu, Wing P. Chan, Chih-Wei Peng

**Affiliations:** 1School of Biomedical Engineering, College of Biomedical Engineering, Taipei Medical University, Taipei 110, Taiwan; 2International Ph.D. Program in Biomedical Engineering, College of Biomedical Engineering, Taipei Medical University, Taipei 110, Taiwan; 3Department of Computer Science and Information Engineering, National Taipei University, New Taipei City 237, Taiwan; 4Department of Physical Medicine and Rehabilitation, School of Medicine, College of Medicine, Taipei Medical University, Taipei 110, Taiwan; 5Department of Physical Medicine and Rehabilitation, Shuang Ho Hospital, Taipei Medical University, New Taipei City 235, Taiwan; 6Department of Physical Medicine and Rehabilitation, Taipei Medical University Hospital, Taipei 110, Taiwan; 7Department of Radiology, Wan Fang Hospital, Taipei Medical University, Taipei 116, Taiwan; 8Department of Radiology, School of Medicine, College of Medicine, Taipei Medical University, Taipei 110, Taiwan; 9School of Gerontology and Long-Term Care, College of Nursing, Taipei Medical University, Taipei 110, Taiwan; 10Research Center of Biomedical Device, Taipei Medical University, Taipei 110, Taiwan

**Keywords:** paired stimulation, repetitive transcranial magnetic stimulation (rTMS), intermittent theta burst stimulation (iTBS), transspinal direct current stimulation (tsDCS), transspinal intermittent theta burst stimulation (ts-iTBS), spinal cord injury (SCI), in vitro stimulation

## Abstract

Paired stimulation of the brain and spinal cord can remodel the central nervous tissue circuitry in an animal model to induce motor neuroplasticity. The effects of simultaneous stimulation vary according to the extent and severity of spinal cord injury. Therefore, our study aimed to determine the significant effects on an incomplete SCI rat brain and spinal cord through 3 min and 20 min stimulations after 4 weeks of intervention. Thirty-three Sprague Dawley rats were classified into six groups: (1) normal, (2) sham, (3) iTBS/tsDCS, (4) iTBS/ts-iTBS, (5) rTMS/tsDCS, and (6) rTMS/ts-iTBS. Paired stimulation of the brain cortex and spinal cord thoracic (T10) level was applied simultaneously for 3–20 min. The motor evoked potential (MEP) and Basso, Beattie, and Bresnahan (BBB) scores were recorded after every week of intervention for four weeks along with wheel training for 20 min. Three-minute stimulation with the iTBS/tsDCS intervention induced a significant (*p* < 0.050 *) increase in MEP after week 2 and week 4 treatments, while 3 min iTBS/ts-iTBS significantly improved MEP (*p* < 0.050 *) only after the week 3 intervention. The 20 min rTMS/ts-iTBS intervention showed a significant change only in post_5 min after week 4. The BBB score also changed significantly in all groups except for the 20 min rTMS/tsDCS intervention. iTBS/tsDCS and rTMS/ts-iTBS interventions induce neuroplasticity in an incomplete SCI animal model by significantly changing electrophysiological (MEP) and locomotion (BBB) outcomes.

## 1. Introduction

Spinal cord injuries (SCIs) alter the motor, sensory, and autonomic functioning of the spinal cord [1]. The effects of SCI depend on the involvement of the spinal cord segment [2]. The severity of the primary injury determines the grade of the patient’s neurologic state, which serves as a prognostic indicator [3]. The majority of SCI patients have stiffness, muscular atrophy, and urine infections, and more than 80% suffer from neuropathic pain [4]. In the US, a greater number of reported incidents of SCI involve the thoracic level [5]. Among the most frequent symptom of thoracic SCI is lower-limb paralysis [6].

SCIs can result in a considerable long-term restructuring of the cerebral cortex [7,8], and complete thoracic SCIs have reduced gray matter volume in the primary motor cortex, which is consistent with atrophy or loss of neurons [9]. After 10–12 weeks following injury, isolated thoracic SCI in rats and mice caused considerable loss of neurons in the hippocampus, cortex, and thalamus, but not at earlier time periods [10,11]. Following SCI, changes in the structure and function of the motor, sensory, and limbic systems above the level of injury have been linked to altered motor [12] and sensory [13] processes as well as neuropathic pain [14]. A series of magnetic resonance imaging (MRI) after acute SCI has revealed immediate, continuous volumetric and diffusivity alterations in these systems [12,13,14,15] above the damage. According to the available evidence, atrophy and clinical impairment are related to a decrease in myelin after a year [12].

A promising therapeutic approach is neuromodulation [16] in the form of repetitive transcranial magnetic stimulation (rTMS), intermittent theta-burst stimulation (iTBS), direct current stimulation (DCS), or paired stimulations [17] of the brain and/or spinal cord, which might enhance neuroplasticity [18] and improve the results of motor rehabilitation in individuals with spinal cord injuries [19]. Magnetic stimulation, rTMS, is a safe and effective non-invasive method to alter the excitability of the brain or spinal cord [20,21]. High-frequency rTMS (≥5 Hz) usually increases corticospinal excitability, whereas low-frequency rTMS (<1 Hz) typically has the opposite effect [20]. In the spinal cord, rTMS causes alterations in presynaptic inhibition in Ia afferent terminals. Hoffman (H)-reflex [22] was decreased with high-frequency rTMS across the motor cortex, but it was increased with low-frequency rTMS [23]. iTBS is a type of rTMS waveform that causes synaptic plasticity in the human cortex [24,25,26] through the postsynaptic mechanism of glutamatergic synapses and the exertion of an excitatory effect on postsynaptic N-methyl-D-aspartate receptors (NMDA) and produces long-term potentiation (LTP) [27]. Transspinal DCS (tsDCS) is another non-invasive electrical stimulation of the spinal cord for the modulation of central nervous system (CNS) excitability [21,28]. According to several earlier investigations, anodal tsDCS is thought to decrease transmission along the ascending spinal pathway [28,29]. The survival of some descending connectivity from supraspinal areas is crucial to the efficiency of spinal stimulation in SCIs, and it is likely associated with coordinated plastic alterations in cortical and spinal connections [30].

Paired stimulation is a brand-new, non-invasive neuromodulation therapy with rarely encountered adverse effects in SCI patients. Through rTMS and tsDCS, it combines simultaneous stimulation of the brain and spinal cord. Additionally, it influences the underlying neuroplastic mechanisms of pre- and postsynaptic neuronal activation, which may benefit patients with SCIs more than conventional therapies such as physiotherapy, occupational therapy, and electrical stimulation [31]. The effectiveness of paired stimulation (iTBS: 1.5 mA; tDCS: 1 mA) was studied in stroke patients utilizing upper extremity functional evaluation, and a significant improvement was noted in Jebsen-Taylor hand function test (JTT) and finger-to-nose test (FNT) scores [32]. In another recent study, it was shown that 20 min of treadmill activity coupled with rTMS (20 Hz, 1800 pulses) on the brain and tsDCS (2.5 mA, 1200 s) on the spinal cord altered corticospinal excitability and increased spinal motor output for up to 30 min in healthy adults [33]. Additionally, after 40 min of paired stimulation (0.1 Hz), one study found a significant decrease in intracortical inhibition and an increase in the facilitation of intracortical and primary motor neurons, which resulted in a reduction in the motor threshold and soleus H-reflex in healthy individuals [34]. One study reported the safety of paired stimulation (iTBS/tsDCS) applied for 30–60 min for up to 4 weeks in an animal brain model without observing any adverse effect on brain tissue biomarkers or histology [35].

Based on the available evidence, this newly investigated paired stimulation approach needs a deeper understanding of its effects due to a lack of focused studies in the literature. Some studies have reported the effectiveness of iTBS/tsDCS or rTMS/tsDCS waveforms in clinical and animal settings [17,32,35,36]. However, no studies have examined different combinations of rTMS, iTBS, and tsDCS applications, particularly for short and long durations, on neuroplasticity after SCI. For this purpose, we tested four different combinations of magnetic and electric stimulation waveforms in an animal model. Therefore, our study aimed to determine the significant effects on an incomplete SCI rat brain and spinal cord through 3 min and 20 min stimulations after 4 weeks of intervention.

## 2. Results

### 2.1. MEP Assessment after Each Week

For the normal group, pre vs. post MEP for pre-week 1 and post-weeks 1, 2, 3, and 4 changed but did not reach the significance level (Figure 1). The sham group showed a change in pre-surgery post_30 min MEP as compared to pre_5 min without achieving a significant level (Figure 2). The 3 min iTBS/tsDCS intervention resulted in a marked change in post-week 2 MEP, particularly in post_5 min (*p* = 0.047 *), 10 min (*p* = 0.029 *), 15 min (*p* = 0.005 *), 20 min (*p* = 0.050 *), and 25 min (*p* = 0.049 *) compared to pre_5 min, as well as in week 4 in post_25 min (*p* = 0.018 *) and post_30 min (*p* = 0.015 *) (Figure 3). In the iTBS/ts-iTBS group, after the 3 min intervention, MEP significantly changed only in post-week 3 in post_5 min (*p* = 0.024 *), post_25 min (*p* = 0.043 *), and post_30 min (*p* = 0.009 *), with a decreasing trend (Figure 4). The 20 min intervention in the rTMS/tsDCS group did not reach the level of significance at any time point, but a varying trend of increase and decrease in MEP values was noted for all weeks (Figure 5). The rTMS/ts-iTBS group after 20 min stimulation had increased MEP values as compared to pre_5 min in weeks 1, 2, and 3, but a significant change was observed during the week 4 intervention in only post_5 min (*p* = 0.047 *) (Figure 6).

### 2.2. Week 1 vs. Week 4 MEP of Each Group

Table 1 compares MEP values between week 1 and week 4 for different time points, particularly pre_5 min of week 1 vs. post_5, 10, 15, 20, 25, and 30 min of week 4, to observe the effects of different stimulations. The following MEP values changed significantly in the 3 min iTBS/tsDCS group in post_25 min (*p* = 0.012 *) and post_30 min (*p* = 0.026 *). The effect size (d) is also shown in Table 1. In contrast, the MEP values for the same time points, i.e., pre_10 min and pre_5 min of week 1 vs. pre_10 min and pre_5 min of week 4 and post_5–30 min of week 1 vs. post_5–30 min of week 4, did not attain a significant level.

### 2.3. MEP before vs. after Surgery and Stimulation Intervention

Figure 7 presents the results of all six groups, including MEP values before SCI induction surgery and after four weeks of stimulation intervention. All of the groups’ pre-surgery MEP values were higher than after surgery, except for the 3 min iTBS/ts-iTBS and 20 min rTMS/tsDCS groups. Only post_10 min (*p* = 0.039 *) in the sham group and pre_5 min (*p* = 0.018 *) in the iTBS/tsDCS group showed significant differences from pre-surgery MEP values. The sham and iTBS/tsDCS groups’ MEP values decreased after surgery and the four-week intervention, while rTMS/tsDCS and rTMS/ts-iTBS groups’ MEP values increased after surgery and the stimulation intervention.

The MEP values before surgery in pre_5 min vs. after surgery and four-week stimulation in post_5–30 min changed without reaching the level of significance. The 3 min iTBS/tsDCS group had significantly changed MEP values after the four-week intervention in post_10 min (*p* = 0.003 *) and post_15 min (*p* = 0.002 *), with an increasing trend (Table 2).

The one-way ANOVA results for MEP time and group factors are presented in Table 3 and Table 4. There was no significant difference observed for the time factor, but the group factor showed significant changes in all weeks.

### 2.4. Basso, Beattie, and Bresnahan (BBB) Locomotor Scale Score

The BBB score changed in all six groups. It changed significantly during week 1 in all groups except for the normal, iTBS/tsDCS, and rTMS/tsDCS groups. During week 2 and week 3, the BBB score improved significantly in all groups except for the normal and rTMS/tsDCS groups. In post-week 4, only the sham and rTMS/ts-iTBS groups reached the level of significance. The change in BBB score is plotted in Figure 8. The week 1 vs. week 4 BBB scores are presented in Figure 9. It only showed a significant change in the rTMS/ts-iTBS group *(p* = *0.002 *)*; other groups’ BBB scores changed as compared to week 1 but did not reach the level of significance. The group factor comparison of the BBB score is presented in Table 5 with the effect size.

## 3. Discussion

In the present study, the effects of paired stimulation on the SCI rat brain and spinal cord were evaluated. In this study, we applied four different paired stimulations on the motor cortex and T10 spinal cord simultaneously in an incomplete SCI animal model. Three-minute stimulation with the iTBS/tsDCS intervention induced a significant (*p* < 0.050 *) increase in MEP after week 2 and week 4 treatments, while 3 min iTBS/ts-iTBS significantly improved MEP (*p* < 0.050 *) only after the week 3 intervention. The 20 min rTMS/ts-iTBS intervention showed a significant change only in post_5 min after week 4.

In our study, iTBS/tsDCS stimulation for 3 min significantly changed MEP in post_5, 10, 15, 20, and 25 min in week 2 and post_25 and 30 min in week 4, respectively (Figure 3). According to a prior study on healthy rats, MEP can be continually increased for up to 30 min following a single session of iTBS [37]. A study on an SCI animal model reported the effects of iTBS stimulation on mild and moderate SCI groups’ neuroplasticity, with enhanced MEPs from baseline after week 1 and week 4 stimulation compared to the severe SCI group [38], which is consistent with our current study results. In our current study, pre- vs. post-week 1 and week 4 also showed a significant increase in post_25 min (*p* = 0.012 *) and post_30 min (*p* = 0.026 *) MEPs (Table 1). In another study conducted on SCI subjects, the iTBS/tsDCS waveform produced a significant increase in MEP (*p* < 0.050 *) [17]. The safety of the iTBS/tDCS waveform ranging from 2.5–4.5 mA/cm^2^ was previously tested in an animal model without any adverse effects on brain tissue biomarkers or scalp tissue histology [35]. A study on stroke patients demonstrated a significant improvement (*p* < 0.050 *) in upper-limb recovery without any side effects after 20 min of 1 mA tDCS and 1.5 mA iTBS application on the primary motor cortex [32].

The 3 min iTBS/ts-iTBS waveform showed a significant change in MEP only in week 3 in post_5, 25, and 30 min. However, the MEP was reduced after the stimulation intervention, inducing inhibitory effects through paired stimulation. A similar effect was observed for the 20 min rTMS/ts-iTBS waveform in post_5 min after week 4. Despite the use of various combinations of iTBS, tsDCS, and rTMS in 3–20 min interventions for four weeks along with wheel training, we did not obtain a lot of significant results for these waveforms. However, iTBS/tsDCS and rTMS/ts-iTBS enhanced cortical excitability through increased MEPs in the SCI animal model. As mentioned in the available literature, paired stimulation can result in instant neuroplasticity in healthy human and animal populations [39]. They reported increased MEP and a decreased spinal threshold for up to 40 min after the rat’s brain and spinal cord had experienced recurrent paired associative stimulation (PAS) for 5–10 min [39].

The changes in BBB scores are presented in Figure 8 and Figure 9. All of the interventions after week 1 to week 4 significantly improved the BBB scores, except for 3 min iTBS/tsDCS and iTBS/ts-iTBS in post-week 4 and for 20 min rTMS/tsDCS in all weeks of the intervention. After surgery, the BBB score was recorded in the range of 3–10 out of 22 in all groups. A greater improvement was noted in the 3 min iTBS/tsDCS group from 0.00 ± 0.00 to 5.00 ± 5.40 (*p* = 0.010 *) after week 2 and to 10.00 ± 9.19 (*p* = 0.340) after week 4. For the 3 min iTBS/ts-iTBS group, week 3 showed an improvement in the BBB score from 5.38 ± 7.11 to 10.13 ± 7.60 (*p* = 0.064), and for the 20 min rTMS/ts-iTBS group, it improved from 3.20 ± 5.54 to 7.80 ± 5.53 (*p* = 0.006 *) after week 2 and to 9.80 ± 5.16 (*p* = 0.008 *) after week 4. Differentiating the effect of these stimulation waveforms is very difficult because the sham group also significantly improved its BBB score. In addition, another study previously mentioned that iTBS does not significantly improve BBB scores in mild or moderate SCI animal models [38].

### 3.1. Strengths of Study

Specific brain or spinal cord stimulations such as rTMS, iTBS, tsDCS, etc., have been previously reported in SCI subjects, but our study explored simultaneous brain and spinal cord stimulations in the form of paired stimulation in an incomplete SCI animal model to investigate the effects on cerebral and spinal cord plasticity through the change in MEP peak-to-peak amplitude and BBB score. In our study, we anesthetized rats before stimulation to record their MEPs before and after the stimulation intervention. We also administered TMS at 120% of RMT for MEP recording, and the stimulation intensity used was 80% of RMT, which is higher than in previous studies. The assessment of MEP through lower-body muscles can also reveal the extent and severity of SCI, which is the key component of clinical rehabilitation.

### 3.2. Limitations of Study

The MEP-based results confirmed the effects on neuroplasticity in the SCI animal model. However, to consider their clinical application, these stimulation waveforms need to be tested on a large sample size. MRI-based studies will be required to highlight the structural or functional changes in cortical and spinal cord circuitry, particularly for incomplete SCI, to develop concrete clinical usage.

### 3.3. Clinical Implications

Spontaneous recovery in terms of a significant increase in MEP amplitude and a change in the BBB score after a 3 min iTBS/tsDCS intervention for four weeks confirmed the effects of paired stimulation in an incomplete SCI animal model, which will have clinical applications in the future for incomplete SCI patients.

## 4. Materials and Methods

### 4.1. Animal Handling

#### 4.1.1. Selection and Care of Animals

A total of thirty-three (*n* = 33) adult Sprague Dawley rats obtained from BioLASCO Taiwan, Yilan, Taiwan, with weights ranging from 300 to 450 g were recruited in the study. Out of 33 rats, 9 were excluded because they died due to urine infections. The remaining twenty-four were divided into six different groups. All of the rats were placed in a clean, temperature- and humidity-controlled environment in a well-equipped animal facility as per ethical standards. They were kept in a 12-h light/12-h dark cycle with easy accessibility to pellet food and water ad libitum. The animals were acclimated for 7 to 10 days. The experimental techniques and animal use/methods were approved by the Institutional Animal Care and Use Committee of Taipei Medical University (IACUC-TMU approval no. LAC-2018-0402, 14 January 2019).

#### 4.1.2. Animal Grouping

All of the animals were divided into six groups: (1) normal (*n* = 4), (2) sham (*n* = 4), (3) iTBS/tsDCS (*n* = 4), (4) iTBS/ts-iTBS (*n* = 4), (5) rTMS/tsDCS (*n* = 3), and (6) rTMS/ts-iTBS (*n* = 5). Groups 3–6 received the stimulation intervention for 3–20 min for 4 weeks (Figure 10).

#### 4.1.3. SCI Surgery

An incomplete SCI was induced using a commercial compression clip (Micro Vascular Clip, RS-6472; Roboz Surgical Instrument Co., Gaithersburg, MD, USA). Each rat was given an intraperitoneal (i.p.) injection of tiletamine-zolazepam (50 mg/kg, i.p.; zoletil, Vibac, Carros, France) and xylazine (10 mg/kg, rompun, Bayer, Leverkusen, Germany). A dorsal laminectomy between T9 and T11 was performed to expose the T10 spinal cord, and a compression clamp was utilized at the T10 level for 60 s through an applicator (Micro Clip Setting Forceps; RS 6496; Roboz Surgical Store, MD, USA). The whole surgical procedure was carried out in an aseptic environment. The rats were checked daily and given postoperative care. The bladder of each rat was squeezed two times a day until spontaneous voiding occurred, and an intramuscular injection of ampicillin (100 mg/kg, subcutaneously (s.c.), Novopharm, Toronto, Canada) was given to prevent infection for five days [38,40] (Figure 11).

### 4.2. Experiment Intervention

#### 4.2.1. MEP Assessment

The rat was positioned in a stereotaxic apparatus following i.p. anesthesia with tiletamine-zolazepam (50 mg/kg, i.p.; zoletil) and xylazine (10 mg/kg, rompun) [37]. To record EMG activity, unilateral monopolar uninsulated 27 G stainless steel needle electrodes (Axon Systems, Hauppauge, NY, USA) were administered into the hindlimb muscle belly of the biceps femoris. The ground electrode was placed in the tail, while the reference electrode was in a paw [41]. Using a Biopac MP-36R four-channel device (Biopac System, Goleta, CA, USA), signals were collected. All MEP stimulations were carried out through a MagVenture MagPro and Cool-40 Rat circular coil (Tonica Electronic, Farum, Denmark). To elicit hindlimb MEPs, the circular coil was set so that it was in slight contact with the scalp surface (locator coordinates AP: ±1.0 mm; ML: ±1.25 mm; bregma as the center) [42]. A detailed method was described in our previous publication [38]. In the SCI rat model, a single TMS pulse elicited MEPs to assess cortical excitability during each session. The resting motor threshold (RMT) is the lowest intensity of TMSs necessary to elicit MEPs from the bicep femoris muscle with at least a 20 µV amplitude in five consecutive trials under muscle relaxation brought on by anesthesia [41]. The reported intensity of TMS for RMT was 100% of the machine output percentage, which corresponds to the highest magnetic field strength [41]. The MEP amplitude was measured two times before intervention stimulation and six times after intervention every five minutes (Figure 12). The recorded MEP comprised 15 single pulses at 10 s intervals with an RMT of 120%. Peak-to-peak MEP amplitude analysis was performed offline [38].

#### 4.2.2. Stimulation Protocol

The stimulation intervention consisted of different combinations of rTMS, iTBS, and tsDCS on the brain and spinal cord in four groups: (1) iTBS/tsDCS comprised iTBS three pulse bursts at 50 Hz repeated at 5 Hz. Each pulse contained a 2 s train, which was repeated every 10 s for 20 repetitions for 600 pulses [26] at 0.05 mA on the brain, while tsDCS was applied at 0.5 mA on the T10 spinal cord for 3 min [17,36]. (2) iTBS/ts-iTBS was applied with the same parameters described earlier for 3 min, with iTBS on the brain and ts-iTBS on the T10 level. (3) rTMS/tsDCS consisted of 20 Hz rTMS on the brain and tsDCS at 0.5 mA on the T10 spinal cord for 20 min. (4) rTMS/ts-iTBS consisted of 20 Hz rTMS on the brain and ts-iTBS on the T10 spinal cord for 20 min. iTBS/tsDCS and iTBS/ts-iTBS were applied for 3 min 3 times a week for 4 weeks, while rTMS/tsDCS and rTMS/ts-iTBS were applied for 20 min once a week for 4 weeks. The rTMS and iTBS stimulation intensity used was 80% of RMT [37]. The sham group protocol was the same as that in the intervention group, but the coil was inverted above the rat’s head [43]. Each animal in each group received 20 min of wheel training after every stimulation intervention.

#### 4.2.3. Basso, Beattie, and Bresnahan (BBB) Scoring

The BBB scale, a locomotion assessment tool to evaluate rat hindlimb function, has a total score of 22 points (normal score, 21; no hindlimb function, 0) and is divided into early (0–7), intermediate (8–14), and late (15–21) stages [42]. For five minutes, we placed a rat inside the track area that we created to observe hindlimb activity. To minimize subjectivity during the assessment, we employed a screening scoring sheet created by Basso [40,44,45,46].

### 4.3. Statistical Analysis

The study data were analyzed in SPSS vers. 17.0 (SPSS, Chicago, IL, USA) with the significance level set to *p* < 0.050 *. All data are presented in the form of mean ± standard deviation (SD) and paired sample t-test for each time factor and group, and one-way ANOVA for time and group factors was utilized for MEP and BBB scores. The figures were prepared using GraphPad Prism 6 software (GraphPad Software, San Diego, CA, USA). Cohen’s effect size (d) is reported for the MEP and BBB score change after 4 weeks of the stimulation intervention as: (1) small = 0.20, (2) medium = 0.50, and (3) large = 0.80.

## 5. Conclusions

The iTBS/tsDCS and rTMS/ts-iTBS interventions induce neuroplasticity in an incomplete SCI animal model by significantly changing electrophysiological (MEP) and locomotion (BBB) outcomes.

## Figures and Tables

**Figure 1 ijms-23-09447-f001:**
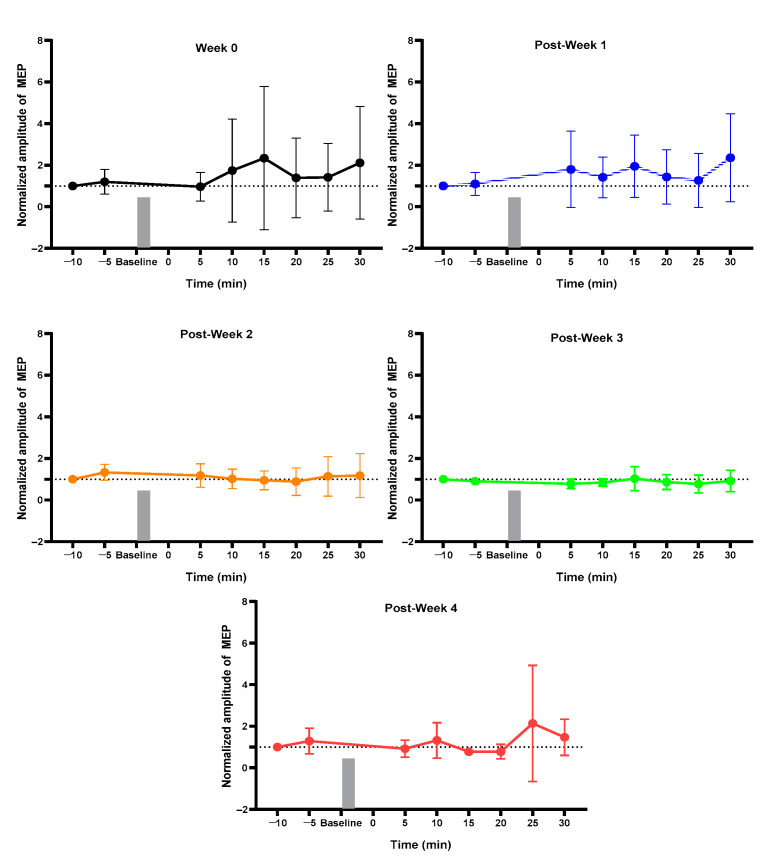
Normal (*n* = 4): Normal group. Week 0, first week; Post-Week 1, MEP after one week of wheel training; Post-Week 2, MEP after two weeks of wheel training; Post-Week 3, MEP after three weeks of wheel training; Post-Week 4, MEP after four weeks of wheel training.

**Figure 2 ijms-23-09447-f002:**
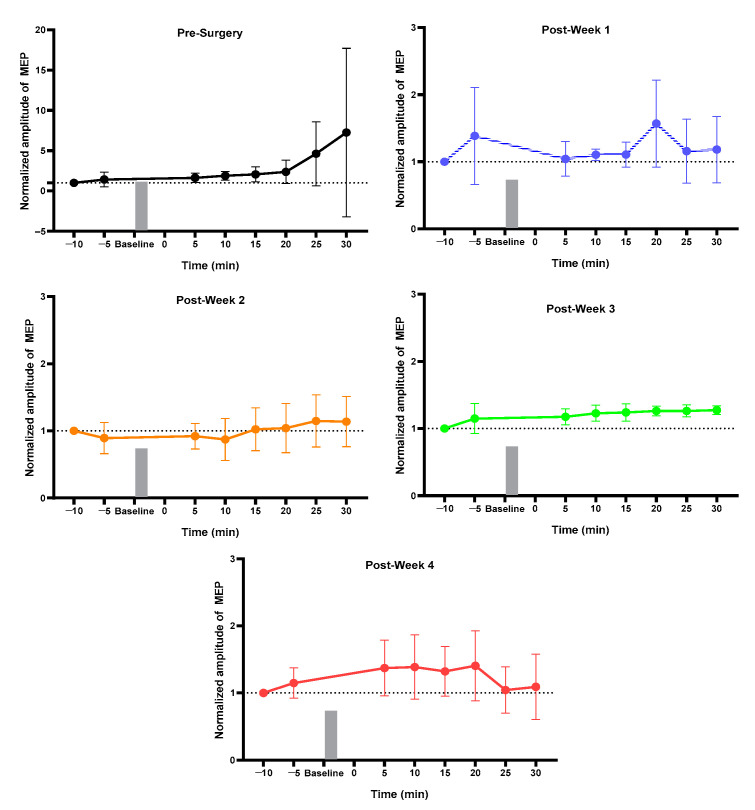
Sham (*n* = 4): Sham group. Pre-Surgery, MEP before surgery; Post-Week 1, MEP after surgery and sham intervention of one week; Post-Week 2, MEP after two weeks of intervention; Post-Week 3, MEP after three weeks of intervention; Post-Week 4, MEP after four weeks of intervention.

**Figure 3 ijms-23-09447-f003:**
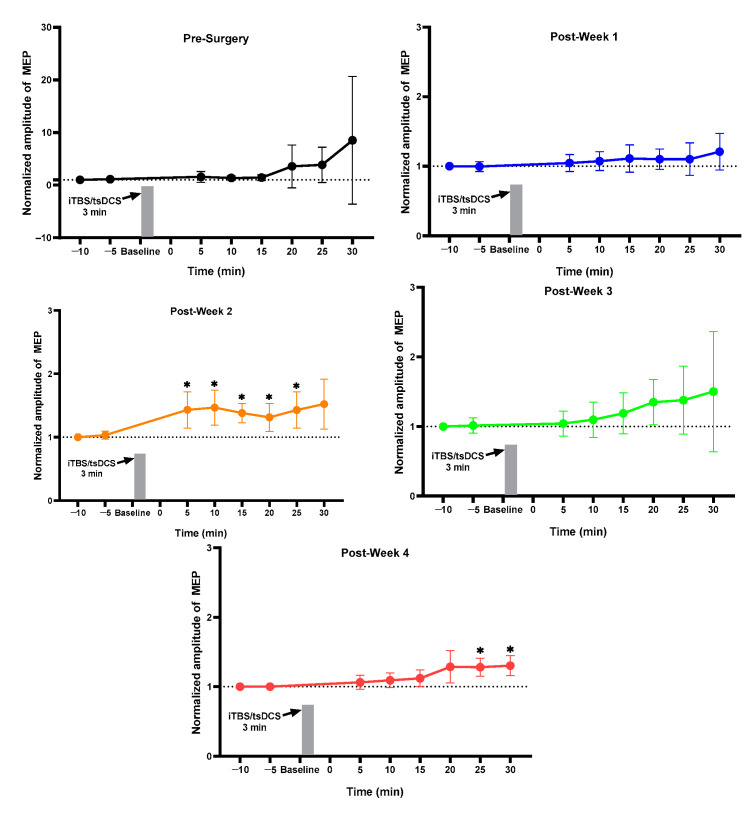
iTBS/tsDCS (*n* = 4). iTBS/tsDCS (3 min). The level of significance was set to *p* < 0.050 *.

**Figure 4 ijms-23-09447-f004:**
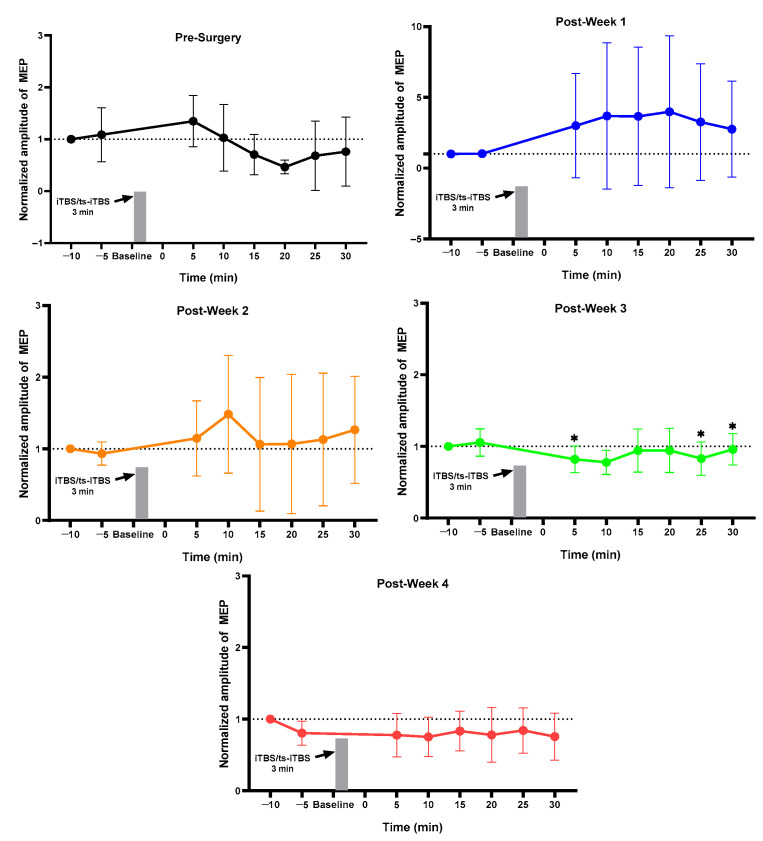
iTBS/ts-iTBS (*n* = 4). iTBS/ts-iTBS (3 min). The level of significance was set to *p* < 0.050 *.

**Figure 5 ijms-23-09447-f005:**
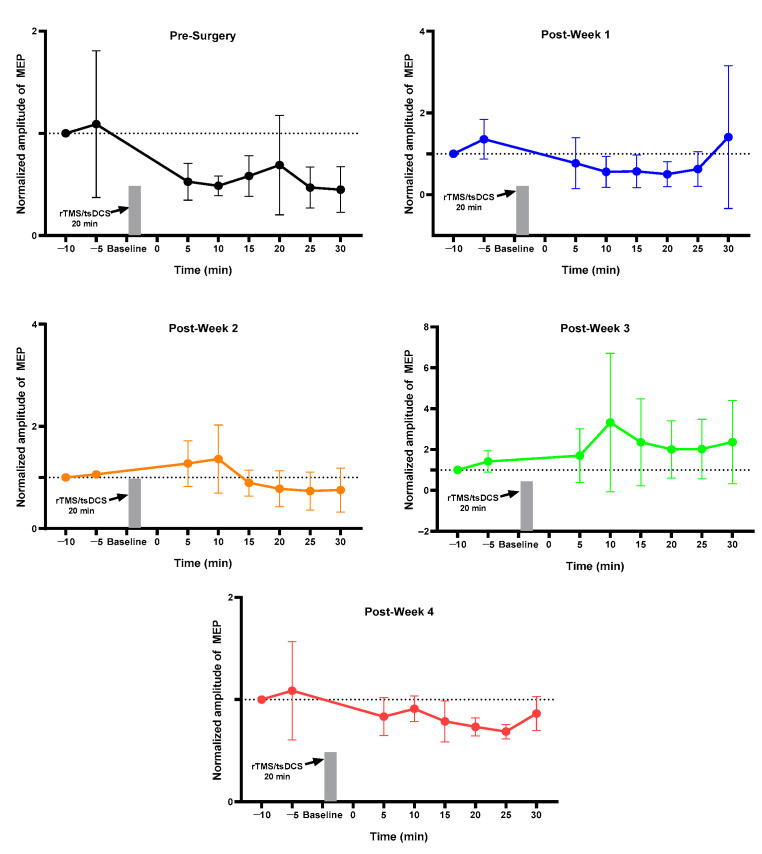
rTMS/tsDCS (*n* = 3). rTMS/tsDCS (20 min).

**Figure 6 ijms-23-09447-f006:**
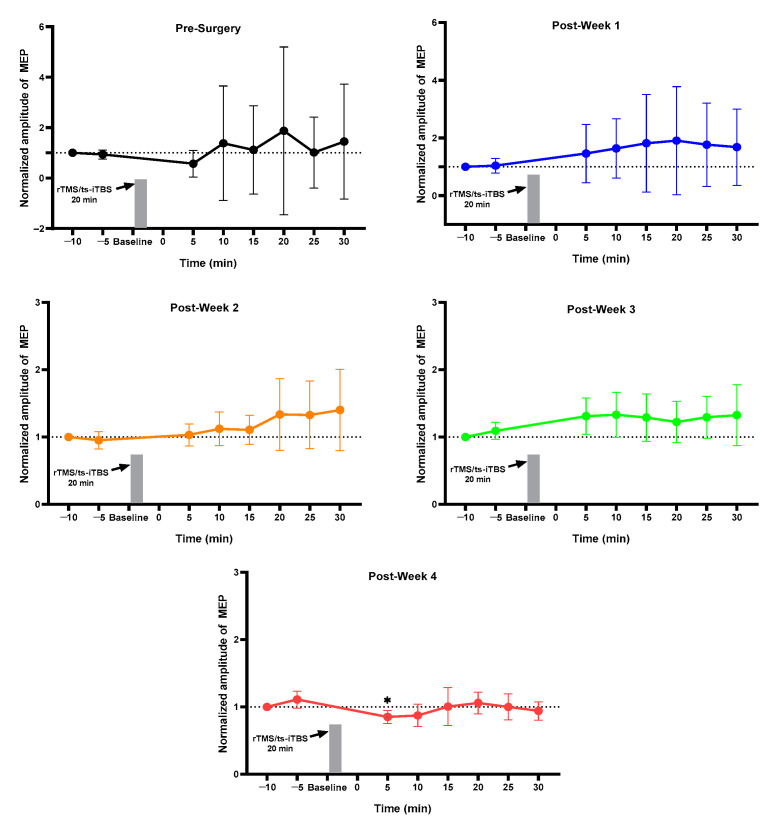
rTMS/ts-iTBS (*n* = 5). rTMS/ts-iTBS (20 min); Pre_5 min vs. post_5, 10, 15, 20, 25, and 30 min were compared using paired sample t-test; The level of significance was set to *p* < 0.050 *.

**Figure 7 ijms-23-09447-f007:**
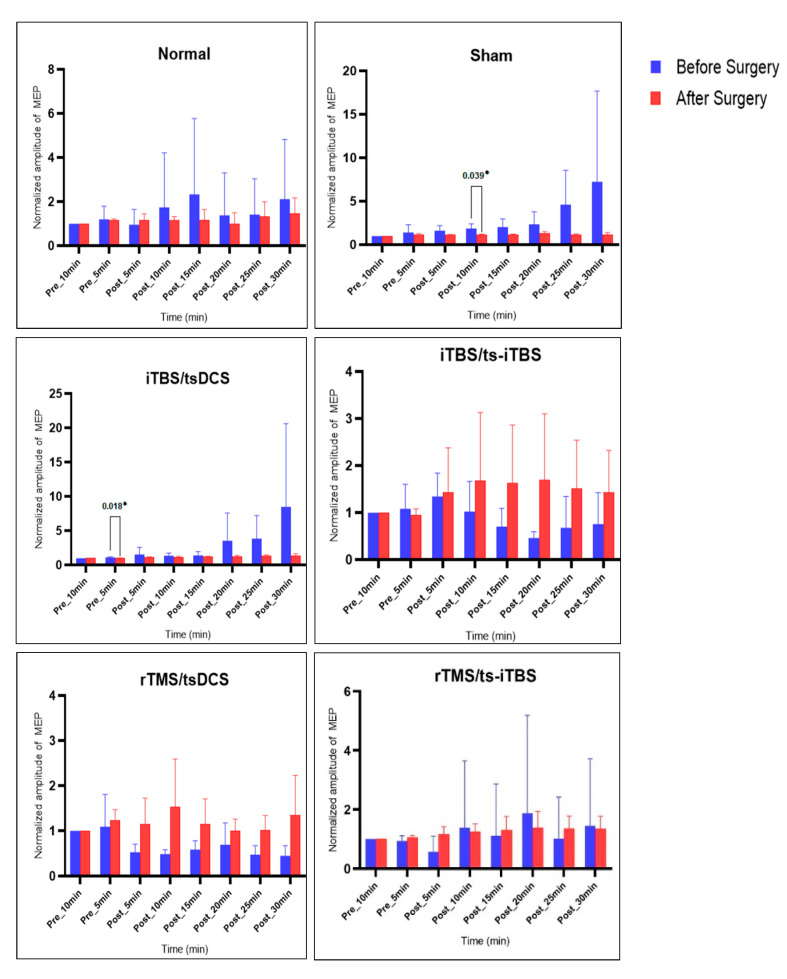
Before Surgery, MEP values before SCI surgery; After Surgery, MEP values after surgery and four-week stimulation intervention. Paired sample *t*-test was used for comparison of before vs. after surgery and stimulation intervention; the level of significance was set to *p* < 0.050 *.

**Figure 8 ijms-23-09447-f008:**
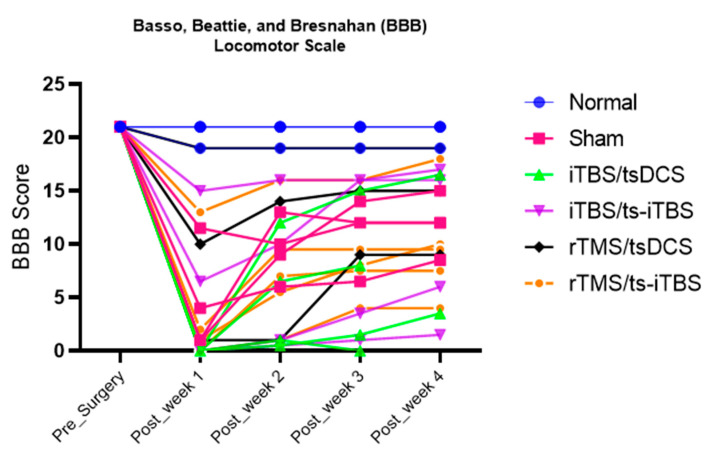
Basso, Beattie, and Bresnahan (BBB) scores for all six groups before surgery and week 1, week 2, week 3, and week 4 after stimulation interventions.

**Figure 9 ijms-23-09447-f009:**
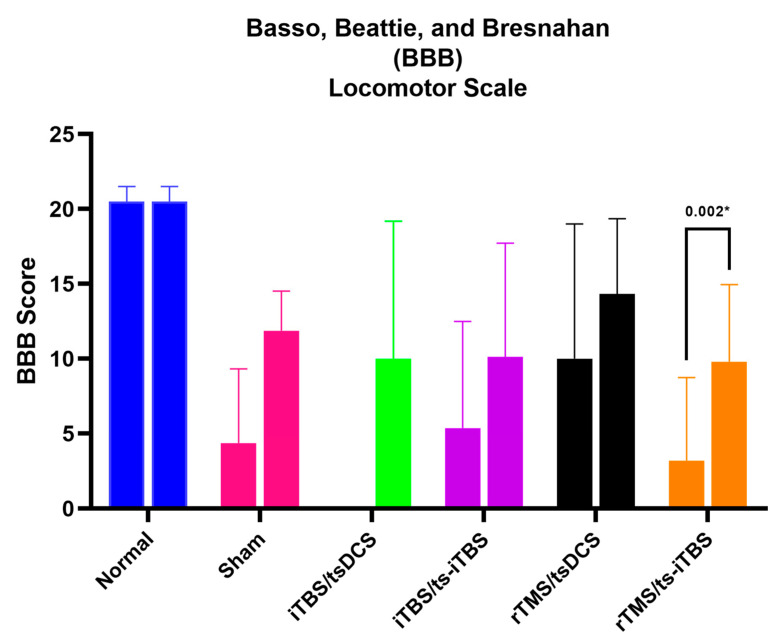
Basso, Beattie, and Bresnahan (BBB) scores for all six groups post-week 1 vs. post-week 4 interventions. d; effect size for sham (−1.22), iTBS/tsDCS (−1.09), iTBS/ts-iTBS (−1.07), rTMS/tsDCS (−1.07), and rTMS/ts-iTBS (−3.23). The level of significance was set to *p* < 0.050 *.

**Figure 10 ijms-23-09447-f010:**
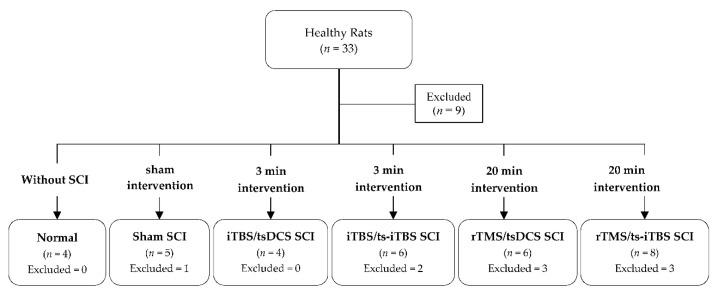
Animal recruitment and grouping for stimulation; SCI, spinal cord injury; iTBS, intermittent theta burst stimulation; tsDCS, transspinal direct current stimulation; ts-iTBS, transspinal intermittent theta burst stimulation; rTMS, repetitive transcranial magnetic stimulation.

**Figure 11 ijms-23-09447-f011:**
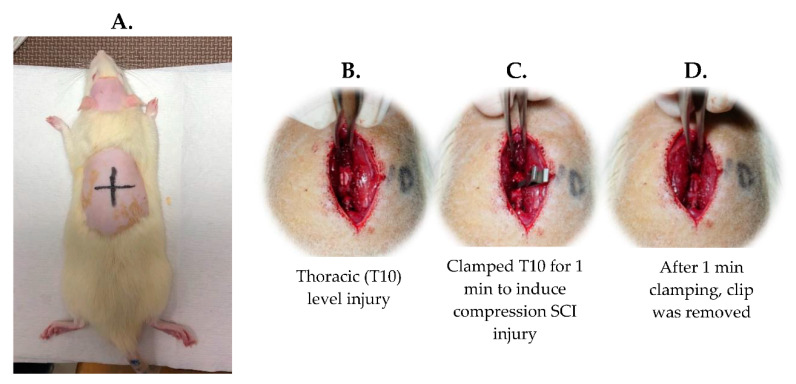
Surgical procedure to induce T10 spinal cord injury. (**A**). Preparation of the rat before the surgery; (**B**). Exposure of thoracic (T10) spinal cord segment; (**C**). T10 level spinal cord was clamped to induce SCI; (**D**). 1 min compression of spinal cord induced compression injury in rat.

**Figure 12 ijms-23-09447-f012:**
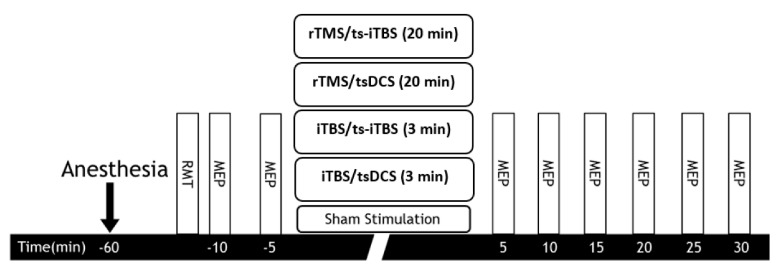
MEP assessment and stimulation interventions for T10 spinal cord injury.

**Table 1 ijms-23-09447-t001:** MEP of both sides in week 1 vs. week 4 for different time points (*n* = 24).

Groups	Normal (*n* = 4)	Sham (*n* = 4)	(iTBS/tsDCS) (*n* = 4)	(iTBS/ts-iTBS) (*n* = 4)	(rTMS/tsDCS) (*n* = 3)	(rTMS/ts-iTBS) (*n* = 5)
**Pre_5 min_w1**	1.10 ± 0.55	1.39 ± 0.72	0.99 ± 0.07	1.02 ± 0.21	1.36 ± 0.48	1.04 ± 0.25
**Post_5 min_w4**	0.92 ± 0.41	1.37 ± 0.41	1.07 ± 0.10	0.77 ± 0.30	0.83 ± 0.18	0.85 ± 0.10
***p*-value**	0.727	0.946	0.393	0.149	0.114	0.066
** *d* **	0.19	0.04	−0.50	0.96	1.56	1.12
**Pre_5 min_w1**	1.10 ± 0.55	1.39 ± 0.72	0.99 ± 0.07	1.02 ± 0.21	1.36 ± 0.48	1.04 ± 0.25
**Post_10 min_w4**	1.32 ± 0.86	1.39 ± 0.48	1.10 ± 0.11	0.75 ± 0.27	0.91 ± 0.13	0.87 ± 0.16
***p*-value**	0.745	0.999	0.279	0.119	0.214	0.058
** *d* **	−0.18	0.00	−0.66	1.08	1.04	1.18
**Pre_5 min_w1**	1.10 ± 0.55	1.39 ± 0.72	0.99 ± 0.07	1.02 ± 0.21	1.36 ± 0.48	1.04 ± 0.25
**Post_15 min_w4**	0.78 ± 0.19	1.32 ± 0.37	1.12 ± 0.12	0.83 ± 0.28	0.78 ± 0.20	1.01 ± 0.28
***p*-value**	0.407	0.826	0.229	0.163	0.230	0.743
** *d* **	0.48	0.12	−0.75	0.92	0.99	0.16
**Pre_5 min_w1**	1.10 ± 0.55	1.39 ± 0.72	0.99 ± 0.07	1.02 ± 0.21	1.36 ± 0.48	1.04 ± 0.25
**Post_20 min_w4**	0.78 ± 0.35	1.41 ± 0.52	1.29 ± 0.23	0.78 ± 0.38	0.73 ± 0.09	1.06 ± 0.16
***p*-value**	0.452	0.952	0.051	0.200	0.172	0.841
** *d* **	0.43	−0.03	−1.58	0.82	1.21	−0.10
**Pre_5 min_w1**	1.10 ± 0.55	1.39 ± 0.72	0.99 ± 0.07	1.02 ± 0.21	1.36 ± 0.48	1.04 ± 0.25
**Post_25 min_w4**	2.14 ± 2.79	1.05 ± 0.35	1.28 ± 0.13	0.84 ± 0.32	0.69 ± 0.07	1.00 ± 0.20
***p*-value**	0.565	0.528	***0.012*** *	0.147	0.156	0.542
** *d* **	−0.32	0.36	−2.74	0.97	1.29	0.30
**Pre_5 min_w1**	1.10 ± 0.55	1.39 ± 0.72	0.99 ± 0.07	1.02 ± 0.21	1.36 ± 0.48	1.04 ± 0.25
**Post_30 min_w4**	1.47 ± 0.87	1.09 ± 0.48	1.30 ± 0.14	0.75 ± 0.33	0.86 ± 0.16	0.94 ± 0.14
***p*-value**	0.622	0.624	***0.026*** *	0.177	0.117	0.309
** *d* **	−0.27	0.27	−2.06	0.88	1.54	0.52

Mean ± SD, mean and standard deviation; the level of significance was set to *p* < 0.050 *; w1, week 1; w4, week 4. Paired sample *t*-test was used to compare pre_5 min in week 1 vs. post_5 min, 10 min, 15 min, 20 min, 25 min, and 30 min in week 4 in each group.

**Table 2 ijms-23-09447-t002:** MEP of both sides before vs. after surgery and stimulation intervention at different time points (*n* = 24).

Groups	Normal (*n* = 4)	Sham (*n* = 4)	(iTBS/tsDCS) (*n* = 4)	(iTBS/ts-iTBS) (*n* = 4)	(rTMS/tsDCS) (*n* = 3)	(rTMS/ts-iTBS) (*n* = 5)
**Pre_5 min**	1.17 ± 0.46	1.20 ± 0.52	1.03 ± 0.08	0.98 ± 0.27	1.20 ± 0.45	1.03 ± 0.17
**Post_5 min**	1.13 ± 0.91	1.23 ± 0.41	1.23 ± 0.49	1.42 ± 1.72	1.02 ± 0.72	1.05 ± 0.58
***p*-value**	0.820	0.803	0.084	0.250	0.306	0.849
**Pre_5 min**	1.17 ± 0.46	1.20 ± 0.52	1.03 ± 0.08	0.98 ± 0.27	1.20 ± 0.45	1.03 ± 0.17
**Post_10 min**	1.23 ± 1.18	1.29 ± 0.47	1.22 ± 0.28	1.55 ± 2.39	1.33 ± 1.70	1.27 ± 1.06
***p*-value**	0.650	0.443	***0.003*** *	0.294	0.760	0.265
**Pre_5 min**	1.17 ± 0.46	1.20 ± 0.52	1.03 ± 0.08	0.98 ± 0.27	1.20 ± 0.45	1.03 ± 0.17
**Post_15 min**	1.41 ± 1.65	1.35 ± 0.57	1.25 ± 0.30	1.44 ± 2.30	1.04 ± 1.08	1.27 ± 1.06
***p*-value**	0.495	0.266	** *0.002* ** ***	0.369	0.517	0.247
**Pre_5 min**	1.17 ± 0.46	1.20 ± 0.52	1.03 ± 0.08	0.98 ± 0.27	1.20 ± 0.45	1.03 ± 0.17
**Post_20 min**	1.07 ± 1.02	1.53 ± 0.82	1.72 ± 1.88	1.45 ± 2.55	0.94 ± 0.81	1.48 ± 1.62
***p*-value**	0.677	0.124	0.111	0.411	0.238	0.169
**Pre_5 min**	1.17 ± 0.46	1.20 ± 0.52	1.03 ± 0.08	0.98 ± 0.27	1.20 ± 0.45	1.03 ± 0.17
**Post_25 min**	1.35 ± 1.52	1.85 ± 2.14	1.81 ± 1.72	1.35 ± 1.97	0.91 ± 0.83	1.28 ± 0.91
***p*-value**	0.587	0.211	0.053	0.402	0.153	0.154
**Pre_5 min**	1.17 ± 0.46	1.20 ± 0.52	1.03 ± 0.08	0.98 ± 0.27	1.20 ± 0.45	1.03 ± 0.17
**Post_30 min**	1.61 ± 1.59	2.39 ± 4.86	2.81 ± 5.65	1.30 ± 1.61	1.17 ± 1.25	1.36 ± 1.15
***p*-value**	0.184	0.294	0.173	0.371	0.902	0.155

Mean ± SD, mean and standard deviation. Paired sample t-test was used to compare pre_5 min before surgery vs. post_5 min, 10 min, 15 min, 20 min, 25 min, and 30 min after surgery and four-week stimulation intervention in each group. The level of significance was set to *p* < 0.050 *.

**Table 3 ijms-23-09447-t003:** MEP comparison using one-way ANOVA with time factor (*n* = 24).

		Sum of Squares	df	Mean Square	F	*p-*Value
Week 0	Between groups	26.68	7	3.81	1.70	0.136
	Within groups	89.58	40	2.24		
Week 1	Between groups	3.32	7	0.47	0.66	0.704
	Within groups	28.74	40	0.72		
Week 2	Between groups	0.29	7	0.04	0.99	0.455
	Within groups	1.66	40	0.04		
Week 3	Between groups	0.95	7	0.14	0.56	0.787
	Within groups	9.80	40	0.25		
Week 4	Between groups	0.18	7	0.03	0.34	0.932
	Within groups	3.09	40	0.08		

Time factor comprises pre_10 min, pre_5 min, post_5 min, post_10 min, post_15 min, post_20 min, post_25 min, and post_30 min; df, degree of freedom.

**Table 4 ijms-23-09447-t004:** MEP comparison using one-way ANOVA with group factor (*n* = 24).

		Sum of Squares	df	Mean Square	F	*p*-Value
Week 0	Between groups	35.34	5	7.07	3.67	** *0.008 ** **
	Within groups	80.93	42	1.93		
Week 1	Between groups	18.93	5	3.79	12.12	** *0.0001* ** * ******* *
	Within groups	13.12	42	0.31		
Week 2	Between groups	0.61	5	0.12	3.85	** *0.006 ** **
	Within groups	1.33	42	0.03		
Week 3	Between groups	6.78	5	1.36	14.31	** *0.0001 **** **
	Within groups	3.98	42	0.01		
Week 4	Between groups	1.28	5	0.26	5.41	** *0.001 *** **
	Within groups	1.99	42	0.05		

Group factor comprises normal, sham, iTBS/tsDCS, iTBS/ts-iTBS, rTMS/tsDCS, and rTMS/ts-iTBS; df, degree of freedom. The level of significance was set to *p* < 0.050 *, *p* < 0.001 **, and *p* < 0.0001 ***.

**Table 5 ijms-23-09447-t005:** BBB score comparison using one-way ANOVA with group factor (*n* = 24).

		Sum of Squares	df	Mean Square	F	*p-*Value
Week 0	Between groups	0.00	5	0.00	--	--
	Within groups	0.00	18	0.00		
Week 1	Between groups	1062.28	5	212.46	7.45	** *0.001 *** **
	Within groups	513.8	18	28.51		
Week 2	Between groups	610.59	5	122.12	3.80	** *0.016 ** **
	Within groups	578.65	18	32.15		
Week 3	Between groups	518.01	5	103.60	3.76	** *0.017 ** **
	Within groups	495.73	18	27.54		
Week 4	Between groups	336.93	5	67.39	2.46	0.078
	Within groups	438.84	16	27.43		

Group factor comprises normal, sham, iTBS/tsDCS, iTBS/ts-iTBS, rTMS/tsDCS, and rTMS/ts-iTBS; df, degree of freedom. The level of significance was set to *p* < 0.050 *, *p* < 0.001 **.

## Data Availability

Not applicable.

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
