# Peer review of "Motor Neuroplastic Effects of a Novel Paired Stimulation Technology in an Incomplete Spinal Cord Injury Animal Model"

_ijms, 2022, doi:10.3390/ijms23169447_

Round 1

Reviewer 1 Report

The Authors used different protocols of paired stimulation on the brain cortex and spinal cord simultaneously over a 4 week period, on rats which had previously undergone experimental partial Spinal Cord Injury (SCI). They tested motor evoked potential (MEP) and Basso, Beattie, and Bresnahan (BBB) scale scores. They find some of the protocols effective in inducing significant changes in both MEP and BBB scores.

Major observations:

1) In order to fully assess the effects of the stimulation, SCI lesion with sham stimulation should be added as a further control group

2) The experimental groups appear too small, especially considering that most results show extremely high standard deviation as compared to the mean values (e.g., 6.13±6.86; 9.13±8.00 etc.; see also Figures 5, 6, 8 and 9 for high variability within the groups)

3) Tables are too long and show little or no significant Results, especially Table 2, should be summarized in the text and deleted, or possibly included as Supplementary material

4) It appears that Fig. 9 substantially replicates Fig. 8: if this is not the case, Authors should explain them better, otherwise they should delete one of the two (Figure 9 preferably, which appears unnecessarily expanded).

5) References: sometimes used inappropriately/not entirely appropriately, see e.g. refs 1-3, lines 54-56

6) General language revision is necessary, e.g., line 97-98 "is investigated in stroke patients ... and observed significant improvement", line 113 "But no one study researched", line 120, "Week vise" (obscure meaning, like Group vise etc); line 129, "after 3 min intervention no one week changed MEP"; "line 221, "One another study"; line 280, "due to urine infection and died"; etc.

Minor points:

- line 48: "magentic"

- line 54: "are based on the involvement" probably means "depend on the involvement" 

- lines 99-109: this part of the Introduction ha too many details (e.g., stimulation frequencies and p values from other studies); they can be moved to the Discussion

Author Response

Response to Reviewer 1 Comments

Major Points

Point 1: In order to fully assess the effects of the stimulation, SCI lesion with sham stimulation should be added as a further control group.

Response to Point 1:

We thank the reviewer for the comment. In this current study, we have included the normal and sham groups already but we agree with the comment and the SCI lesion with sham stimulation as a further control group will be considered for future research.

This is the first study that employed four different combinations of paired stimulations in an animal SCI model. There are some limitations of this study, as many groups were included, and the sample size of the study was diluted. The study involved a one-month duration of paired stimulation with daily stimulation interventions which was quite difficult to continue with large sample size. This was the pilot trial for testing different stimulation parameters to understand their effects in the SCI model for translation into the clinical setting. The effect size is also mentioned for MEP and BBB score change after 4 weeks of stimulation intervention in table 1 and figure 9 of the revised manuscript.

We have added the limitation in the discussion in the revised manuscript (Limitations, page 14, Lines 276-279). Based on these limitations, the results of this study will help to conduct research on a large sample size in the future.

“The MEP-based results confirmed the effects on the neuroplasticity in the SCI animal model. But to consider the clinical application, these stimulation waveforms need to be tested on a large sample size.” (Limitations, page 14, Lines 276-279).

Point 2: The experimental groups appear too small, especially considering that most results show extremely high standard deviation as compared to the mean values (e.g., 6.13±6.86; 9.13±8.00 etc.; see also Figures 5, 6, 8 and 9 for high variability within the groups).

Response to Point 2:

We thank the reviewer for the comment. We agree with the comment that the sample size is small for each group, but this is the first study that employed four different combinations of paired stimulations in an animal SCI model to test and evaluate magnetic and electric stimulations simultaneously which is the main contribution of this study. Because of the many groups, the sample size of the study was diluted. The study involved a one-month duration of paired stimulation with daily stimulation interventions which was quite difficult to continue with large sample size. The effect size is also mentioned for MEP and BBB score change after a 4-week stimulation intervention in table 1 and figure 9 of the manuscript. But we agree that in the future, we must include more rats in each group to reduce the variability in mean values and get consistent data so the results of this study will help to understand the paired stimulation effects in the SCI model for translation into the clinical setting.

Point 3: Tables are too long and show little or no significant Results, especially Table 2, should be summarized in the text and deleted, or possibly included as Supplementary material.

Response to Point 3:

We thank the reviewer for the comment. We agree with the comment and Table 2 is summarized in the result section and deleted from the manuscript. (Results, page. 5-6,7, lines. 149-152).

Point 4: It appears that Fig. 9 substantially replicates Fig. 8: if this is not the case, Authors should explain them better, otherwise they should delete one of the two (Figure 9 preferably, which appears unnecessarily expanded).

Response to Point 4:

We thank the reviewer for the comment. We agree with the comment and Fig. 9 is removed from the manuscript as it is explaining the Fig. 8 details for each group. (Results, page 12).

Point 5: References: sometimes used inappropriately/not entirely appropriately, see e.g. refs 1-3, lines 54-56.

Response to Point 5:

We thank the reviewer for the comment. We have checked the references and cited their information to support the spinal cord injury and its extent in our manuscript. (Introduction, page. 2, line. 55-57).

References

  1. Krassioukov A. Autonomic function following cervical spinal cord injury. Respiratory physiology & neurobiology. 2009;169(2):157-64.
  2. Fakhoury M. Spinal cord injury: overview of experimental approaches used to restore locomotor activity. Reviews in the neurosciences. 2015;26(4):397-405.
  3. Oyinbo CA. Secondary injury mechanisms in traumatic spinal cord injury: a nugget of this multiply cascade. Acta neurobiologiae experimentalis. 2011;71(2):281-99.

Point 6: General language revision is necessary, e.g., line 97-98 "is investigated in stroke patients ... and observed significant improvement", line 113 "But no one study researched", line 120, "Week vise" (obscure meaning, like Group vise etc); line 129, "after 3 min intervention no one week changed MEP"; "line 221, "One another study"; line 280, "due to urine infection and died"; etc.

Response to Point 6:

We thank the reviewer for the comment. We agree with the comment and the general language revisions are made for lines 97-98, lines 113, 120, 129, 221, and 280 in the manuscript in the highlighted part. 

Minor Points:

Point 1: - line 48: "magentic".

Response to Point 1:

We thank the reviewer for the comment. We agree with the comment and the spelling is revised to “magnetic”.

Point 2: line 54: "are based on the involvement" probably means "depend on the involvement".

Response to Point 2:

We thank the reviewer for the comment. We agree with the comment and revised it to "depend on the involvement".

Point 3: lines 99-109: this part of the Introduction ha too many details (e.g., stimulation frequencies and p values from other studies); they can be moved to the Discussion.

Response to Point 3:

We thank the reviewer for the comment. We agree with the comment and some of the values are moved to the Discussion.

Reviewer 2 Report

Thank you for inviting me to review this manuscript entitled ‘Motor Neuroplastic Effects by the Novel Paired Stimulation Technology in an Incomplete Spinal Cord Injury Animal Model’. This study aims to investigate the effect of a different combination of 4-weeks pairs stimulation on the MEP and locomotion function in rats. Results of the study showed that the iTBS/tsDCS and rTMS/ts-ITBS are superior to other kinds of pair stimulations in inducing neuroplasticity. The finding of the study could be important for developing a pair stimulation protocol for SCI patients.

In general, the paper is easy to follow, the authors reviewed the relevant literature. The methodology has been described in detail. While I am confused about the presentation of results and, from my point of view, the results have not been discussed thoroughly.

My major concerns are:

1.           My feeling is that several statistical comparisons have been performed and reported in the results. However, the authors did not discuss these results. For example, the results are presented in table 2. The authors should try to suggest why there is no significant difference observed.

2.           Moreover, the results of new statistical comparisons were mentioned in the discussion section. Thus, the discussion section is filled with information that should be presented in the results section. For example, in 243-249. If I understand correctly, the results presented here were the comparisons between post-surgery week1 and post-surgery week2,3,4. While the results presented in Table 4 were the comparisons between pre-surgery and post-surgery weeks 1,2,3,4. The discontinuity impaired the readability of the manuscript.

3.           If the changes in BBB score in the sham group and pair-stimulation groups is similar, the author should try to explain why the changes of MEP didn’t translate into functional changes.

4.           The authors should avoid recapping the information and statistical results that have already been presented in the result section in the discussion section. (Line 219 – 223)

5.           Some statistical comparisons with a p value greater than 0.05 were highlighted and reported as statistically significant eg. Line 220 (p = 0.051*), table 3 iTBS/tsD8CS, post_25 min p=0.053*. I think this mistake may have a great impact on the overall interpretation of the results.

Other comments:

6.           No effect size is provided for the statistical comparisons

7.           The rationale for selecting 20min and 3 min for the corresponding pair stimulations has not been mentioned.

8.           The authors described that ANOVA was used, but the results of ANOVA have not been reported anywhere in the manuscript.

Author Response

Response to Reviewer 2 Comments

Major Points

Point 1: My feeling is that several statistical comparisons have been performed and reported in the results. However, the authors did not discuss these results. For example, the results are presented in table 2. The authors should try to suggest why there is no significant difference observed.

Response to Point 1:

We thank the reviewer for the comment. We agree with the comment and table 2 is removed from the manuscript as per the other reviewer’s suggestion. The possible reason for the non-significant results was the small sample size for each group. We have added the limitation in the discussion in the revised manuscript (Limitations, page 14, Lines 276-279). However, this is the first study that employed four different combinations of paired stimulations in an animal SCI model to test and evaluate magnetic and electric stimulations simultaneously which is the main contribution of this study. Because of the many groups, the sample size of the study was diluted. The study involved a one-month duration of paired stimulation with daily stimulation interventions which was quite difficult to continue with large sample size. The effect size is also mentioned for MEP and BBB score change after 4 weeks of stimulation intervention in table 1 and figure 9 of the manuscript. But in the future, we must include more rats in each group to reduce the variability in mean values and get consistent data so the results of this study will help to understand the paired stimulation effects in the SCI model for translation into the clinical settings.

Point 2: Moreover, the results of new statistical comparisons were mentioned in the discussion section. Thus, the discussion section is filled with information that should be presented in the results section. For example, in 243-249. If I understand correctly, the results presented here were the comparisons between post-surgery week1 and post-surgery week2,3,4. While the results presented in Table 4 were the comparisons between pre-surgery and post-surgery weeks 1,2,3,4. The discontinuity impaired the readability of the manuscript.

Response to Point 2:

We thank the reviewer for the comment. We agree with the comment and table 4 is removed from the manuscript and replaced by figure 9 to show a comparison between post-week 1 and post-week 4 BBB scores along with the effect size. (Results, page 12).

Point 3: If the changes in BBB score in the sham group and pairstimulation groups is similar, the author should try to explain why the changes of MEP didn’t translate into functional changes.

Response to Point 3:

We thank the reviewer for the comment. Even though the BBB score change was significant but their changing trend was different in sham and paired stimulation groups which may be due to the 20 min wheel training for each group. But the change in BBB score in paired stimulation groups is also obvious. It is difficult to explain the possible effect of MEP translation into a functional change due to the limited sample size. But we will consider the reviewer’s comment for future research and include a large sample size to explore this effect.  

Based on week vise analysis, the BBB score changed significantly during week 1 for all the groups except normal, iTBS/tsDCS, and rTMS/tsDCS groups. During week 2 and week 3, BBB score improved significantly for all groups except normal and rTMS/tsDCS groups. While post week 4, only sham and rTMS/ts-iTBS groups attained the level of significance.

While based on the week 1 vs. week 4 comparison, only the rTMS/ts-iTBS group reached the significant level after 4 weeks of intervention as shown in figure 9 in the result section. (Results, page 12).

Point 4: The authors should avoid recapping the information and statistical results that have already been presented in the result section in the discussion section. (Line 219 – 223).

Response to Point 4:

We thank the reviewer for the comment. We agree with the comment and lines 219-223 in the discussion section are removed and revised. (Discussion, page 13, lines 230-231).

Point 5: Some statistical comparisons with a p value greater than 0.05 were highlighted and reported as statistically significant eg. Line 220 (p = 0.051*), table 3 iTBS/tsD8CS, post_25 min p=0.053*. I think this mistake may have a great impact on the overall interpretation of the results.

Response to Point 5:

We thank the reviewer for the comment. We agree with the comment and the following: line 220 (p = 0.051*), table 3 iTBS/tsDCS, and post_25 min p=0.053* are revised in the manuscript.

Point 6: No effect size is provided for the statistical comparisons.

Response to Point 6:

We thank the reviewer for the comment. We agree with the comment and the effect size for statistical comparisons is added to table 1 for MEP and figure 9 for BBB score, respectively in the manuscript. (Results, Page 5,6, 12) and (Methods, Page 16, Lines, 369-370).

Point 7: The rationale for selecting 20min and 3 min for the corresponding pair stimulations has not been mentioned.

Response to Point 7:

We thank the reviewer for the comment. We tried to mimic the paired stimulation for 3 min iTBS/tsDCS used clinically to examine the effects of stimulation. Based on the available literature, the previous study used 3 min paired stimulation for clinical application in chronic spinal cord injury patients. The following reference is cited in the objective part of the manuscript:

References:

  1. Effects of paired stimulation with specific waveforms on cortical and spinal plasticity in subjects with a chronic spinal cord injury

While another study examined the adverse effects of tDCS/iTBS stimulation waveform in a rat model for up to 20 mins without observing any adverse effects and is cited in the objective part of the manuscript.

References:

  1. Safety of Special Waveform of Transcranial Electrical Stimulation (TES) in vivo Assessment

“Some of the studies reported the effectiveness of iTBS/tsDCS or rTMS/tsDCS waveforms in clinical and animal settings [17, 32, 35, 36]. But no one study examined the different combinations of rTMS, iTBS, and tsDCS applications particularly short and long durations on neuroplasticity after SCI”.(Introduction, page 3, lines 114-117).

Point 8: The authors described that ANOVA was used, but the results of ANOVA have not been reported anywhere in the manuscript.

Response to Point 8:

We thank the reviewer for the comment. We agree with the comment and ANOVA results are presented in tables 3, 4, and 5 in the result section. (Results, pages 9,10,12).

Round 2

Reviewer 2 Report

The authors have addressed all my comments, or provided a justification if they are unable to address. Although there are shorting comings on the study methodology (eg, small sample size), I am satisfied with the manuscript.